# The Terminal Extensions of Dbp7 Influence Growth and 60S Ribosomal Subunit Biogenesis in *Saccharomyces cerevisiae*

**DOI:** 10.3390/ijms24043460

**Published:** 2023-02-09

**Authors:** Julia Contreras, Óscar Ruiz-Blanco, Carine Dominique, Odile Humbert, Yves Henry, Anthony K. Henras, Jesús de la Cruz, Eduardo Villalobo

**Affiliations:** 1Instituto de Biomedicina de Sevilla, Hospital Universitario Virgen del Rocío/CSIC/Universidad de Sevilla, 41013 Seville, Spain; 2Departamento de Genética, Facultad de Biología, Universidad de Sevilla, 41004 Seville, Spain; 3Molecular, Cellular and Developmental Biology Unit (MCD), Centre de Biologie Intégrative (CBI), Université de Toulouse, CNRS, UPS, 31000 Toulouse, France; 4Departamento de Microbiología, Facultad de Biología, Universidad de Sevilla, 41004 Seville, Spain

**Keywords:** Dbp7, DEAD-box protein, RNA helicase, *Saccharomyces cerevisiae*, ribosome, 60S ribosomal subunit, ribosome assembly factor

## Abstract

Ribosome synthesis is a complex process that involves a large set of protein *trans*-acting factors, among them DEx(D/H)-box helicases. These are enzymes that carry out remodelling activities onto RNAs by hydrolysing ATP. The nucleolar DEGD-box protein Dbp7 is required for the biogenesis of large 60S ribosomal subunits. Recently, we have shown that Dbp7 is an RNA helicase that regulates the dynamic base-pairing between the snR190 small nucleolar RNA and the precursors of the ribosomal RNA within early pre-60S ribosomal particles. As the rest of DEx(D/H)-box proteins, Dbp7 has a modular organization formed by a helicase core region, which contains conserved motifs, and variable, non-conserved N- and C-terminal extensions. The role of these extensions remains unknown. Herein, we show that the N-terminal domain of Dbp7 is necessary for efficient nuclear import of the protein. Indeed, a basic bipartite nuclear localization signal (NLS) could be identified in its N-terminal domain. Removal of this putative NLS impairs, but does not abolish, Dbp7 nuclear import. Both N- and C-terminal domains are required for normal growth and 60S ribosomal subunit synthesis. Furthermore, we have studied the role of these domains in the association of Dbp7 with pre-ribosomal particles. Altogether, our results show that the N- and C-terminal domains of Dbp7 are important for the optimal function of this protein during ribosome biogenesis.

## 1. Introduction

The ribosome is one of the most complex machines that any cell synthesises and assembles [1]. Ribosomes are made of two subunits, the large ribosomal subunit (r-subunit) and the small r-subunit. In the yeast *Saccharomyces cerevisiae*, as in other eukaryotes, these subunits are also referred to as the 60S and 40S r-subunits, respectively. In this yeast, the ribosome synthesis process implicates 4 different ribosomal RNA (rRNA) species (18S for the 40S r-subunit and 25S, 5.8S, and 5S for the 60S r-subunit), 79 ribosomal proteins (r-proteins; 33 for the 40S r-subunit and 46 for the 60S r-subunit), about 75 small nucleolar RNAs (snoRNAs), and more than 250 protein *trans*-acting factors, also known as assembly factors (AFs) [2,3].

RNA helicases of the DEx(D/H)-box protein family and related families (e.g., Ski2-like family) represent the largest class of AFs involved in ribosome biogenesis. Indeed, up to 21 RNA helicases have been described to participate in this process in *S. cerevisiae*, 20 of them in the biogenesis of cytoplasmic ribosomes and 1 in the biogenesis of mitoribosomes [4,5,6]. RNA helicases are motor proteins that use the energy of ATP hydrolysis to exert work onto RNA [7,8,9]. In vitro, RNA helicases have been found to perform a wide range of biochemical activities. Some of these enzymes are able to catalyse ATP-dependent double-stranded RNA (dsRNA) unwinding, hence the name given to this family of proteins. Other activities include RNA clamping, dsRNA destabilization, RNA annealing, and protein displacement [10,11,12,13,14,15,16,17]. Thus, RNA helicases act as enzymes able to bind and remodel RNA or ribonucleoprotein particles (RNPs) in an NTP-dependent manner [18,19]; consequently, RNA helicases are found ubiquitously in nature, participating in virtually all aspects of cellular RNA metabolism [20,21]: notably transcription, pre-mRNA splicing, RNA editing, RNA interference, RNA export, small RNA biogenesis, ribosome biogenesis, translation, RNA degradation, and RNA quality control mechanisms (e.g., [11,22]).

RNA helicases of the DEx(D/H)-box protein family are members of the superfamily 2 (SF2); superfamilies are classified according to the occurrence and characteristics of various conserved motifs in their primary sequence [8,23,24]. Among the conserved motifs, all RNA helicases harbour the Walker A and B motifs [25], required for NTP binding and hydrolysis, as well as other motifs required for nucleic acid binding and for coupling NTP hydrolysis to helicase activity (e.g., [8,9,16]). From structural analyses, it becomes evident that the motifs are a signature for a structurally conserved helicase core region, which consists of two tandem RecA-like domains connected by a flexible short linker [4,9,14]. In all cases, the core region is flanked by specific N- and C-terminal extensions of variable lengths, which could confer physiological specificity (e.g., recruitment to distinct target RNPs, subcellular localisation, platform for interaction of cofactors, etc.) or provide complementary catalytic activities to RNA helicases [8,16,26]. Figure 1 outlines the primary sequence of Dbp7, the RNA helicase of the DExD-box protein family studied in this work, and highlights the different conserved motifs of the core region along with the sequence, length, and characteristics of its non-conserved N- and C-terminal extensions.

Terminal extensions of RNA helicases, especially at the N-terminus, have few predicted structures or consist of intrinsically disordered domains, and are poorly characterized. In any case, it has been suggested that sequence elements within the extensions help to modulate the functional activities of the proteins, either directly or through interactions with other factors [27]. In *S. cerevisiae*, the non-conserved C-terminal region of Dhh1 is required for recruitment of the helicase to specific mRNAs to promote their decay [28]; the N-terminus of Prp16 is essential for its nuclear localisation and for its interaction with specific partners on the spliceosome, among which is another RNA helicase called Brr2 [29,30]. The N- and C-terminal domains of Mss116 contribute to its RNA chaperone function, at least in part by mediating interactions with RNA and RNP substrates [31,32]; the C-terminal extension of Prp22 has been described as a binding platform for interaction partners and to regulate its ATPase activity [33]. The N- and C-terminal domains of Dhr1 have important functions in ribosome biogenesis in vivo [34]. The structure of unique C-terminal domains of RNA helicases has been determined for Prp43 and Mtr4, which have been demonstrated to function in RNA binding, RNA-dependent ATPase activity, and interactions with specific adaptor proteins [35,36].

Dbp7 is a non-essential DEGD-box protein involved in the synthesis of 60S r-subunits [37]. Recently, Dbp7, which has RNA-dependent ATPase activity, has been shown to regulate the dynamic base-paring between the snoRNP chaperone snR190 and the pre-rRNA within early pre-60S particles [38,39]. In this work, we report on the functional relevance of the specific N- and C-terminal extensions of Dbp7 in relation to its function during 60S r-subunit biogenesis. We show that the N-terminal extension of Dbp7 is responsible for targeting the protein to the nucleus, in part via a standard bipartite nuclear localization signal (NLS). Consistently, we found that removal of this NLS slightly affected growth and ribosome maturation, while the complete deletion of Dbp7 N-terminal extension restrained growth and led to abnormal 60S r-subunit biogenesis. Likewise, we have made truncations in the C-terminal extension, which were also detrimental for cell growth and 60S r-subunit biogenesis. Importantly, both the N-terminal and C-terminal extensions of Dbp7 seem to be important, but not essential, for the association of the protein with pre-ribosomal particles. In conclusion, the truncation of the N- or C-terminal extension of Dbp7 underscores the relative importance of these regions of the protein for its functions during ribosome biogenesis.

## 2. Results

### 2.1. Truncation of the N- and C-Terminal Extensions of Yeast Dbp7

Yeast Dbp7 is a quasi-essential DEGD-box protein involved in rRNA remodelling on the earliest pre-60S r-particles [37,38,39,40]. While the ATPase activity of Dbp7 is clearly important for its function during ribosome biogenesis [38,39], the role of its terminal extensions remains unknown. A search for similarities of these extensions in other proteins led us to conclude that the N-terminal extension has no homologies to other DEx(D/H)-box or non-related proteins, and apparently shows a disordered architecture. In contrast, the C-terminal extension shows some sort of ordered structure, as well as a DUF4217 domain of unknown function (from T638 to G703) which is present in the C-terminus of different RNA helicases, among which are yeast Has1, Dbp4 and Spb4 [41]. In agreement with this, the N-terminal part of Dbp7 shows a high probability of being intrinsically disordered, as predicted by both the Protein DisOrder prediction System (PrDOS) [42] and the DisEMBL^TM^ tool [43]. Moreover, the Dbp7 structure predicted in the AlphaFold Protein Structure Database [44,45] features a mostly unstructured N-terminal extension of Dbp7, while the C-terminal extension comprises several alpha helices (see Appendix A).

To address the importance of these extensions for the biological function of Dbp7, we generated a series of progressive and independent truncations in the N-terminal and C-terminal extensions that flank the helicase core, positioned between the amino acid 163, just upstream of the Q-motif, and the amino acid 559, just downstream of motif VI of Dbp7 (Figure 1). Different truncation alleles were generated by site-directed mutagenesis on a HA-tagged *DBP7* ORF and cloned into the YCplac22 *CEN TRP1* plasmid; then, their functions were tested in vivo in a *dbp7∆* strain using the plasmid shuffle procedure.

We obtained the following *dbp7* mutants (see Figure 1 and Appendix A for details), all of which were expressed under the control of the cognate *DBP7* promoter: (1) HA-*dbp7∆N10*, an allele harbouring a deletion of the codons corresponding to the first 10 amino acid residues; (2) HA-*dbp7∆N162*, an allele harbouring a deletion of codons corresponding to the first 162 amino acid residues from the N-terminus; (3) HA-*dbp7∆NLS*, an allele expressing a HA-tagged Dbp7 variant lacking 31 amino acids residues (V48 to S78) of the N-terminus; (4) HA-*dbp7∆C694-742*, an allele harbouring a deletion of the codons corresponding to the last 49 amino acids; and (5) HA-*dbp7∆C636-742*, an allele harbouring a deletion of the codons corresponding to the last 107 amino acids. All of these alleles also encode an N-terminal double HA tag to allow for the detection of the different truncated proteins. As a control, we used a wild-type *DBP7* allele also preceded by a N-terminal double HA tag, which has been previously proven to support wild-type growth [37].

### 2.2. The N-Terminal Extension of Dbp7 Is Required for Efficient Cell Growth and for Production of 60S r-Subunits

To address the in vivo function of the N-terminal extension of Dbp7, we first examined the growth phenotype of the truncated Dbp7-expressing strains on selective minimal SD-Trp plates. As shown in Figure 2A, deletion of the first N-terminal 10 amino acids of Dbp7 (HA-*dbp7∆N10*) had only a mild effect on cell growth as compared to the wild-type counterpart, suggesting that this region is mostly dispensable under laboratory conditions. Elimination of the complete N-terminal extension of Dbp7 (HA-*dbp7∆N162*) resulted in a slow growth similar to that caused by the absence of the Dbp7 protein. Consistently, doubling times of ca. 1.5, 1.9, 5.8, and 5.8 h were obtained in liquid SD-Trp medium at 30 °C for the HA-*DBP7*, HA-*dbp7∆N10*, HA-*dbp7∆N162,* and *dbp7∆* strains, respectively. As shown in Figure 2B and Appendix A, both N-terminal truncated forms of Dbp7 were detected by Western blotting, although they accumulated to a lesser extent than wild-type HA-Dbp7, especially the HA-Dbp7∆N162 variant protein.

We then examined the consequences of truncating the N-terminal extension of Dbp7 for ribosome biogenesis. To do so, we performed polysome and r-subunit profile analyses. As shown in Figure 3, the HA-*dbp7∆N10* strain displayed a wild-type polysome profile; however, the HA-*dbp7∆N162* mutant exhibited a mild reduction in the amount of free 60S r-subunits relative to free 40S r-subunits, a reduction in polysomes, and an accumulation of half-mer polysomes. Half-mer polysomes correspond to mRNAs associated with integral numbers of ribosomes *plus* a stalled 48S preinitiation complex at the initiation codon [46,47]. This profile was very similar to that obtained with an isogenic *dbp7∆* strain, suggesting that the HA-*dbp7∆N162* allele impairs 60S r-subunit production. To confirm this, we quantified total r-subunits by using run-off and low-Mg^2+^ sucrose gradients. We observed no significant reduction in the total 60S/40S r-subunit ratio in the HA-*dbp7∆N10* strain, but a significant reduction of about 25% in the HA-*dbp7∆N162* strain. This deficit was similar to that observed for the *dbp7∆* strain. Altogether, we conclude that the N-terminal extension of Dbp7 is required for cell growth and optimal 60S r-subunit biogenesis, the impairments caused by the lack of the N-terminal extension being of the same magnitude as those resulting from the complete absence of the Dbp7 protein.

### 2.3. The N-Terminal Region of Dbp7 Contains a Functional Nuclear Localization Signal

Dbp7 is a nucleolar *trans*-acting factor [37]. To explore the amino acid sequence of Dbp7 for signals targeting the protein to the nucle(ol)us, we used independent prediction methods, such as cNLS Mapper [48], ELM [49], or SeqNLS [50]. These methods predicted both monopartite and bipartite nuclear localization signals (NLSs), with significant high scores only within the N-terminal extension of Dbp7.

We selected the region of 31 amino acids contained between residues V48 and S78, which all programs predicted to be part of a putative NLS of Dbp7. To experimentally validate whether or not the N-terminal extension of Dbp7 contained a bona fide NLS, we cloned the sequence corresponding to the first N-terminal 162 amino acids of Dbp7 into a *CEN LEU2* plasmid (pADH111-(GA)_5_-3xyEGFP, see Appendix A). In the resulting plasmid, the N-terminal sequence of Dbp7 was fused in frame via a (GA)_5_ decapeptide linker, with that corresponding to three tandem GFP monomers (3xyEGFP) at the C-terminus (construct hereafter named Dbp7.N162); the fusion construct was expressed from the strong *ADH1* promoter. As a functional control, we fused to the 3xyEGFP reporter a variant of the latter sequence, which lacked the sequence corresponding to the selected 31 amino acids (from V48 to S78; hereafter named Dbp7.N162-∆NLS); this fusion was also cloned under the control of the *ADH1* promoter in plasmid pADH111, as described above. As a positive control for nuclear targeting, we used the NLS of the SV40 large T-antigen fused similarly to 3xyEGFP. As a negative, primarily cytoplasmic control, we used a construct expressing the (GA)_5_-3xyEGFP alone. A triple rather than a single yEGFP fusion was used to increase the fluorescence intensity and to ensure dependence on an NLS-mediated transport, thus minimising the passive entry to the nucleus of the different constructs. Each construct was transformed into YKL500, a wild-type strain expressing the nucleolar marker protein Nop58-yEmCherry from its genomic locus. Transformants were grown on specific selective SD media to the mid-log phase, whole cell extracts were prepared for Western blot analysis of the expression levels of the constructs, and living cells were observed by fluorescence microscopy. As shown in Appendix A, the proteins corresponding to the different constructs were fairly well expressed in this strain. As shown in Figure 4, the Dbp7.N162-(GA)_5_-3xyEGFP construct displayed mainly a nucleoplasmic localization. This result indicates that the N-terminal domain of Dbp7 is sufficient to drive the nuclear import of the 3xyEGFP reporter, similarly to the NLS of the SV40 large T-antigen. The Dbp7.N162-∆NLS-(GA)_5_-3xyEGFP construct was inefficiently directed to the nucleus since the signal remained mostly cytoplasmic, similarly to the signal observed for (GA)_5_-3xyEGFP alone. Altogether, these experiments suggest that the N-terminal sequence of Dbp7, from V48 to S78, which was predicted to include a putative NLS, is indeed part of a bona fide NLS able to target the 3xyEGFP reporter to the nucleoplasm. However, the nucleolar accumulation of Dbp7 might depend on another part of the protein in the helicase core domain or the C-terminal extension.

### 2.4. The NLS (V48 to S78 Sequence) of Dbp7 Is Required for Optimal Growth but Its Deletion Does Not Affect Polysome Profiles

To explore the functional relevance of the NLS present between V48 and S78 residues in Dbp7, we constructed an HA-tagged Dbp7 variant lacking this specific sequence (hereafter named HA-Dbp7∆NLS), which was cloned in YCplac22 and expressed from the cognate promoter of *DBP7* in a *dbp7∆* strain. The protein was detected by Western blotting at levels similar to those of a full-length wild-type HA-Dbp7 (Appendix A). Then, we studied the growth phenotype of the HA-*dbp7∆NLS* mutant. A very minor increase in the doubling time in liquid cultures was observed (1.7 h for the HA-*dbp7∆NLS* strain compared to the 1.5 h for the wild-type HA-*DBP7* strain, Figure 5A). Moreover, this strain showed wild-type polysome profiles (Figure 5B). We conclude that Dbp7 does not exclusively rely on the V48 to S78 sequence, predicted to contain an NLS, for nuclear import, as deletion of this sequence does not affect polysome profiles under laboratory conditions and leads only to a very minor effect on cell growth.

### 2.5. The N-Terminal Region of Dbp7 Is Necessary for Efficient Nuclear Import

To further define the Dbp7 sequences within its N-terminal extension required for nuclear import, we fused different Dbp7 protein variants in frame with the triple yEGFP protein via a (GA)_5_ decapeptide linker at their C-terminal end and expressed the resulting constructs from plasmids into the YKL500 strain. Then, the expression of the variants was analysed by Western blotting and their subcellular localisation by fluorescence microscopy. The selected Dbp7 protein variants were the following: a Dbp7 variant protein lacking the V48 to S78 sequence from its N-terminal extension (Dbp7∆NLS) and a Dbp7 variant protein lacking its complete N-terminal extension (Dbp7∆N162). Cells expressing the (GA)_5_-3xyEGFP-tagged wild-type Dbp7 and the (GA)_5_-3xyEGFP reporter alone were included in all analyses as controls. All proteins were expressed, as shown by Western blot analysis (Appendix A). When the subcellular localization of the Dbp7 protein variants was assessed (see Figure 6), we found that the full-length Dbp7-(GA)_5_-3xyEGFP construct was strongly enriched, as expected, in the nucle(ol)us. However, the Dbp7∆NLS-(GA)_5_-3xyEGFP was detected in both the nucleus and the cytoplasm, and some residual nucleolar localization was observed since the brightest nuclear GFP signals colocalised with the nucleoli stained with Nop58-yEmcherry. Interestingly, the Dbp7∆N162-(GA)_5_-3xyEGFP was detected mainly in the cytoplasm, although some minor nuclear signal could also be observed. Altogether, our results suggest that the N-terminal extension of Dbp7 harbours the sequences required for efficient nuclear import; however, the rest of the protein also contains sequences allowing modest nuclear targeting capacity. Whether this is occurring directly or through a “piggy-back” import process must be clarified.

### 2.6. The C-Terminal Extension of Dbp7 Is also Required for Efficient Cell Growth and for Production of 60S r-Subunits

We also decided to define the function of the C-terminal domain of Dbp7. We thus performed two terminal deletions, HA-*dbp7∆C694-742* and HA-*dbp7∆C636-742* (Figure 1B), both compromising the putative function of the DUF4217 domain present in the C-terminal extension of Dbp7 and some other RNA helicases. As shown in Figure 2A, both deletions significantly impaired cell growth—strikingly, in a more pronounced manner than the complete loss of the Dbp7 protein. Doubling times of 7.3 and 6.7 h were obtained for the HA-*dbp7∆C694-742* and HA-*dbp7∆C636-742,* respectively, compared to the 1.5 h and 5.8 h obtained for the isogenic HA-*DBP7* and *dbp7∆* strains, respectively, in liquid SD-Trp medium at 30 °C. No dominant negative phenotype was detected when the HA-*dbp7∆C694-742* and HA-*dbp7∆C636-742* alleles were expressed from the *CEN* YCplac22 plasmids in a wild-type strain (Appendix A). Moreover, Western blot analysis of the C-terminal deletion mutants showed that while the HA-Dbp7∆C694-742 protein was minimally expressed, the levels of HA-Dbp7∆C636-742 protein were similar to those of the wild-type Dbp7 protein (Figure 2B and Appendix A).

Polysome analyses were performed with cell extracts prepared from the HA-*dbp7∆C694-742* and HA-*dbp7∆C636-742* mutants and the isogenic wild-type strain grown in SD-Trp at 30 °C. As shown in Figure 3, these mutations led to a significant shortage of 60S r-subunits, as revealed by the deficit of free 60S *versus* 40S r-subunits and the appearance of half-mer polysomes. This shortage was further confirmed by quantification of total r-subunits in run-off low Mg^2+^ sucrose gradients. An A_254_ 60S/40S ratio of approximately 1.5 was calculated for the HA-*dbp7∆C694-742* and HA-*dbp7∆C636-742* mutants compared to a ratio of 1.9 for the isogenic wild-type HA-*DBP7* strain (reduction of ca. 20%). Together, these data demonstrate that the C-terminal extension of Dbp7 is, as is its N-terminal extension, strictly required for normal cell growth and 60S r-subunit synthesis.

### 2.7. Importance of Terminal Extensions of Dbp7 in Pre-rRNA Processing

To further analyse the role of the terminal extensions and the NLS of Dbp7 in ribosome biogenesis, we studied whether the corresponding mutants showed defects in pre-rRNA processing. To do so, the HA-*dbp7∆NLS*, HA-*dbp7∆N10*, HA-*dbp7∆N162,* and HA-*dbp7∆C636-742* mutants, as well as the isogenic HA-*DBP7* and *dbp7∆* strains, were grown in SD-Trp at 30 °C. Total RNA was extracted and the levels of pre-rRNA intermediates were assayed by Northern blotting. As previously reported [37,38,39], an accumulation of 35S and 23S pre-rRNAs were detected in the *dbp7∆* strain (Figure 7, lane 1). Moreover, the steady-state levels of 32S, 20S, and especially those of 27SB pre-rRNAs were clearly reduced, which likely accounts for the net deficit in 60S r-subunits observed in this strain. Consistently with the absence of apparent defects in growth and ribosome production, expression of the HA-Dbp7∆N10 protein led to wild-type levels of pre-rRNAs. Expression of the HA-Dbp7∆NLS protein induced minor pre-rRNA processing defects, including a slight accumulation of 27SA_2_ pre-rRNA (Figure 7, lanes 2–4), which are likely not sufficient to alter the polysome profiles observed to be unaffected in Figure 5. Expression of the HA-Dbp7∆C636-742 protein resulted in pre-rRNA processing defects similar to those observed in the *dbp7∆* strain (Figure 7, lane 6); however, expression of the HA-Dbp7∆N162 protein led to strong pre-rRNA processing defects that were different from those of the *dbp7∆* strain. These consisted of a strong accumulation of 35S and 27SA_2_ pre-rRNAs and reduced levels of 27SB pre-rRNAs, but no apparent decrease in 20S pre-rRNA levels (Figure 7, lane 5). Together, these results suggest that while the removal of a few amino acids in the N-terminal extension or in the NLS of Dbp7 minimally affects 60S r-subunit maturation, a long C-terminal truncation leads to defects similar to the complete lack of the Dbp7 protein. Interestingly, a long N-terminal truncation of Dbp7 impairs 60S r-subunit maturation in a different manner, suggesting that the HA-Dbp7∆N162 protein could differently interact with pre-60S r-particles.

### 2.8. Influence of the N- or C-Terminal Extensions of Dbp7 on Its Association with Pre-Ribosomal Particles

Having demonstrated the requirement of the N- and C-terminal extensions of Dbp7 for cell growth and 60S r-subunit biogenesis, we further explored whether the basis of the cell growth and ribosome synthesis defects of the N- and C-terminal truncated mutants was due to an impairment of the recruitment of the mutated variant proteins to pre-60S r-particles. To this end, we first analysed the sedimentation behaviour of wild-type and mutant Dbp7 proteins in sucrose gradients. Total extracts were prepared from wild-type cells and all truncated *dbp7* mutants grown in SD-Trp at 30 °C, and then subjected to low Mg^2+^ sucrose gradient ultracentrifugation. Then, fractions were collected and analysed by Western blotting. As shown in Figure 8, an extended peak of wild-type HA-Dbp7 was found to be associated with high-molecular-mass complexes that sediment at positions practically similar to those of mature 60S r-subunits, which were identified by the signal of the 60S r-subunit protein uL3. These complexes most likely represent pre-60S r-particles. Consistently, the *trans*-acting factor Erb1, which was demonstrated to bind to pre-60S r-particles (e.g., [51,52]), also sedimented with high-molecular-mass complexes, some of them co-sedimenting with those containing Dbp7 (Figure 8, upper panel, lanes 10 and 11). Some wild-type Dbp7 proteins sediment in lighter fractions. The significance of this remains unclear, but clearly does not correspond to free protein. We favour the hypothesis that this reflects a heterogeneity of the Dbp7-containing pre-60S r-particles or sub-complexes thereof. Interestingly, when we examined the distribution of HA-Dbp7∆N10, HA-Dbp7∆N162, and HA-Dbp7∆C636-742 in sucrose gradients, we could not find any difference in the sedimentation behaviour of the Dbp7 variant proteins in the gradients nor in that of the control proteins (Figure 7, second, third, and bottom panels). As expected, very weak signals were detected when examining the HA-Dbp7∆C694-742 variant protein (Figure 7, fourth panel).

To further study the association of the mutant Dbp7 proteins with pre-60S r-particles, we next performed anti-HA immunoprecipitation experiments and studied the precipitation efficiency of 35S and 27S pre-rRNAs by Northern blotting. As shown in Figure 9A, the HA-Dbp7∆N10 and HA-Dbp7∆NLS proteins were as efficiently immunoprecipitated as the wild-type HA-Dbp7 control by the anti-HA resin. However, the HA-Dbp7∆N162 and HA-Dbp7∆C636-742 proteins were only poorly immunoprecipitated, although mild levels of both of them could be detected. Moreover, when pre-rRNAs were analysed (Figure 9B), both HA-Dbp7∆N10 and HA-Dbp7∆NLS purifications could clearly co-enrich 35S and 27S pre-rRNAs identically to the HA-Dbp7 control. These data indicate that Dbp7 binds at an early stage during the formation of pre-60S r-particles and remains associated with 27SB-containing particles. However, neither HA-Dbp7∆N162 nor HA-Dbp7∆C636-742 proteins were able to precipitate those pre-rRNAs above background levels, indicating that they were not stably associated with pre-ribosomal particles in the immunoprecipitation conditions.

Altogether, these findings suggest that truncation of either the N- or C-terminal extension of Dbp7 does not substantially decrease the association of Dbp7 with pre-60S r-particles when studied by sedimentation experiments through sucrose gradients. However, both the ∆N162 and the ∆C636-742 truncations clearly impair the association with these particles in immunoprecipitation experiments. Since these later experiments impose harsher physicochemical constraints on the molecular interactions compared to sucrose gradients (e.g., higher salt concentration, presence of detergents, physical pulldown), we propose that the ∆N162 and ∆C636-742 truncations change or weaken the association of Dbp7 with pre-ribosomal particles, resulting in a decreased affinity in immunoprecipitation conditions. Still, the fact that, at least by sedimentation experiments, all variant proteins (except HA-Dbp7∆C694-742) appeared in high-molecular-mass complexes confirmed the previous observation that these proteins are still able to enter the nucleus. Clearly, more experiments are required to completely clarify this issue.

## 3. Discussion

It has been shown that Dbp7 is a non-essential DExD-box protein required for 60S r-subunit biogenesis [37]. Through its ATP hydrolysis, Dbp7 induces the release of the snR190 snoRNP chaperone from early pre-60S r-particles, thus allowing the maturation of these particles into 60S r-subunits [38,39]. Consistent with this role, the K197A mutation in Dbp7 conserved motif I, which has been demonstrated to abolish in vitro ATP hydrolysis and to mildly affect growth, but clearly less drastically than the *dbp7∆* mutation [39]. Paradoxically, both the *dbp7*[K197A] and the *dbp7∆* mutations comparably inhibit 60S r-subunit production [39]. Mutations in conserved motifs II (E308Q) and VI (R553A) lead to similar growth and ribosome synthesis defects, demonstrating the importance of the conserved motifs for Dbp7 function ([38] and J.C., unpublished results). A detailed functional analysis of the conserved sequence motifs has been previously carried out for different yeast ATP-dependent helicases of this SF2 class, including members participating in ribosome biogenesis (e.g., [40,53,54,55,56,57,58,59]). In this work, we have studied the functional importance of the non-conserved N- and C-terminal domains of Dbp7 for cell viability, the subcellular localization of the protein, and ribosome biogenesis. In distinct DEx(D/H)-box proteins, these extensions have been described to have roles as platforms for protein and/or RNA binding to provide functional specificity (e.g., [29,58,60,61]), although, in general, the role of these extensions is poorly understood.

Our analysis shows that a small deletion in the N-terminal extension of Dbp7 is tolerated, leading to almost no apparent effects on cell growth and ribosome synthesis. In contrast, deletion of the complete N-terminal region leads to similar growth and 60S r-subunit production defects as the *dbp7* null allele, but results in pre-rRNA processing defects that are different from those observed in the absence of Dbp7. By Western blotting analyses, we observed stable expression of the truncations, although the levels of these constructs were lower than those of the wild-type Dbp7 protein. This indicates that the entire N-terminal region is required for the in vivo function of Dbp7 and contributes to its stability. This region displays no particular features or domains except the presence of a putative bipartite NLS, which we demonstrated serves the nuclear import of Dbp7. Our results show that the deletion of this putative NLS has a mild effect on growth and pre-rRNA processing, consistent with the fact that the Dbp7∆NLS protein is inefficiently imported into the nucleus. These defects are, however, not strong enough to translate into detectable changes in polysome profiles. As observed in our microscopy analysis, it is likely that enough Dbp7∆NLS protein still reaches the nucleolus (Figure 6, bright GFP foci co-localising with Nop58-yEmCherry), indicating that at least one additional NLS must be present in the primary sequence of Dbp7. This sequence may also lie in the N-terminal extension of Dbp7, as suggested by our results, which showed that a Dbp7 construct that is completely deprived of this extension is mainly localised in the cytoplasm. Many other examples of redundant NLS have been reported (e.g., [62,63]). Still, we cannot rule out an event of piggybacking for nuclear import, as reported for the human RNA helicase DDX6 [64]. The role of the N-terminal extension of other yeast RNA helicases has been previously explored; however, to the best of our knowledge, this is the first report that has specifically addressed the functional role of N- and C- terminal extensions of an RNA helicase involved in ribosome biogenesis, besides the work by Roychowdhury et al. on Dhr1 [34]. In the particular cases of RNA helicases involved in splicing, a large portion of the unique N-terminal extensions of these proteins can be deleted without significantly altering the function of the proteins in vivo. For example, the deletion of the N-terminal extension of Prp2 has been described as having no measurable effect in vivo [65]; similarly, the deletion of most residues (90 out of 118) of the N-terminal domain of Prp43 did not affect its function [66,67]. The deletion of half of the N-terminal extension of Prp16 (deletion of the first 150 residues) causes no defect, although further deletion of this domain results in a growth defect (deletion of the first 228 residues) or becomes lethal (deletion of the first 300 residues) [29]. Most importantly, it has been shown that the N-terminal domain of Prp16 is required for nuclear localization of the protein and for mediating specific interactions with the spliceosome [29]. In another example, it was shown that ca. 350 residues out of the 505 residues of the unique N-terminal extension of Prp22, including a S1 motif involved in spliceosome binding, can be deleted without consequences for cell growth if expressed from an episomal plasmid [61]. Further deletion of its N-terminal domain clearly affects the biological function of Prp22 and fails to sustain cell growth, unless the N-terminal region is provided in *trans* [61]. Similarly, while the complete N-terminal extension (ca. 500 residues) of Brr2 is required for cell viability and efficient splicing, partial truncations of this extension cause specific defects during the rearrangements that involve Brr2 in splicing [68].

We have also analysed the C-terminal extension of Dbp7. This extension has a disordered structure with no recognizable features except that described by SMART [41] as the DUF4217 domain, a short domain found in the C-terminal extension of many RNA helicase proteins and whose function is unknown. Our analysis shows that the C-terminal domain is essential for Dbp7 function; strikingly, expression of the two alleles with C-terminal truncations led to growth defects even more severe than the complete deletion of the *DBP7* gene. This perhaps occurs due to the fact that the C-terminal part of Dbp7, which is structured in contrast to the N-terminal region, contributes to the architecture of the RecA-2 domain. However, neither truncation showed a dominant negative phenotype over wild-type Dbp7. In Dhr1, an RNA helicase involved in ribosome biogenesis whose C-terminal extension has been studied, this domain is required for cell growth and 40S r-subunit biogenesis [34]. In other RNA helicases, such as Prp2, Prp16, Prp22, and Prp43, the C-terminal extensions are also essential; strikingly, the C-terminal domains of these RNA helicases show some sequence conservation among themselves on most of their ca. 300 residues [29,61,69]. In contrast, the unique, non-conserved C-terminal extension of Dhh1, an RNA helicase involved in the decay of specific mRNAs, is not needed for cell proliferation [28]. In the case of Mss116, an RNA helicase required for mitochondrial splicing, the C-terminal extension is required for its mitochondrial functions as it facilitates the ATPase and strand displacement activities of the protein [31].

As discussed above, N- and C-terminal extensions of some RNA helicases have been described to modulate their activities, either directly or through the interaction with other factors. Moreover, they can regulate their association with the macromolecular complexes in which they exert their functions (i.e., spliceosome). For example, the C-terminal extension of Dhr1 has been shown to be required for the dissociation of the protein from 90S pre-ribosomal particles and for the interaction with its co-factor Utp14 [34]. In order to rationalize the loss-of-function phenotype of the truncations in the N- and C-terminal extensions of Dbp7, we analysed pre-rRNA processing in the different mutants. We showed that while the longest C-terminal truncation mutant from this study leads to defects similar to those of the *dbp7* null strain, the mutant that lacks the entire N-terminal domain of Dbp7 leads to a different phenotype, suggesting that Dbp7 could have more than one specific role during the biogenesis of 60S r-subunits. We also studied whether these terminal extensions were involved in the association of Dbp7 with pre-ribosomal particles. Our sedimentation experiments strongly suggested that neither the binding to nor the release of Dbp7 from pre-ribosomal particles seemed to be affected by the truncations. In contrast, co-IP experiments showed that the association with pre-60S particles of Dbp7 variants bearing the ∆N162 or ∆C636-742 truncation was substantially weakened. We believe that the terminal extensions of Dbp7 impair the stable recruitment of the RNA helicase into pre-ribosomal particles, which is only revealed experimentally in the harsh physicochemical conditions of immunoprecipitation experiments. The fact that the HA-*dbp7∆N162* mutant leads to a pre-rRNA processing defect different to that of the *dbp7* null strain is consistent with this scenario; thus, binding of the HA-Dbp7∆N162 protein to pre-60S r-particles might interfere differently with 60S r-subunit maturation than the absence of Dbp7.

In summary, we have expanded the knowledge regarding Dbp7 function by showing that (i) the N-terminal extension of Dbp7 is required for its in vivo role and harbours at least one functional NLS; (ii) the C-terminal domain of Dbp7 is also required for ribosome biogenesis in vivo; and (iii) the N- or C-terminal extensions play an important role in the association of Dbp7 with pre-ribosomal particles. We have previously shown that Dbp7 is involved in a functional interaction network comprising the uL3 r-protein and a set of other ribosome assembly factors, including the Npa1-subcomplex (Dbp6, Npa1, Npa2, Nop8, Rsa3) [70,71]. Recently, we have shown that Dbp7 promotes the release of snR190 from early pre-ribosomal particles [39]. Further studies on Dbp7 are needed to define whether or not its non-conserved extensions are involved in these specific functions.

## 4. Materials and Methods

### 4.1. Strains and Plasmids

The relevant genotypes of the *S. cerevisiae* strains used in this study are listed in Appendix A. Unless otherwise indicated, the strains were derivatives of strain W303 [72]. Plasmids used in this study are listed in the Appendix A. Information regarding the construction of the different plasmids and the oligonucleotides that we used to generate them will be available upon request. *Escherichia coli* XL1-Blue was used for all recombinant DNA techniques.

### 4.2. Media and Culturing

Yeast cells were routinely grown at 30 °C in YPD (1% yeast extract, 2% peptone, 2% glucose) medium or in SD (0.15% yeast nitrogen base, 0.5% ammonium sulphate, 2% glucose) broth supplemented with the appropriate amino acids and bases as nutritional requirements [73]. All solid media contained 2% agar. Yeast was transformed using the lithium acetate method [74]. Bacteria were routinely grown in LB (0.5% yeast extract, 1% tryptone, 0.5% NaCl) broth with or without agar, in the presence of 100 µg/mL ampicillin, when required.

To test the in vivo function of the different truncated *dbp7* alleles, a *dbp7∆* strain harbouring plasmid YCplac33-DBP7 (*CEN URA3 DBP7*) was transformed with YCplac22 (*CEN TRP1*) plasmids carrying various *dbp7* truncated alleles (see Appendix A). Trp^+^ transformants were selected and streaked out on plates containing 5-fluoroorotic acid (5-FOA) to counter-select for the *URA3* plasmid [75]. Clones growing on these plates were recovered on fresh SD-Trp plates, and their growth was then assessed on YPD and selective SD-Trp media.

### 4.3. Nucleic Acid Manipulations

DNA amplifications were carried out by PCR. Site-directed mutagenesis was performed following the previously described methodology [76] and using DNA constructions ligated in plasmid templates.

### 4.4. Protein Extraction and Western Blotting Analyses

Total yeast protein extracts were prepared and analysed by western blotting according to standard procedures [77]. Different mouse monoclonal antibodies, at the dilutions recommended by the manufacturers, were used: anti-HA (Roche, Basel Switzerland), anti-Pgk1 (Invitrogen, Schwerte, Germany), and anti-GFP (Roche, Basel Switzerland). The Nhp2 protein was detected using a specific rabbit polyclonal antibody [78]. Secondary goat anti-mouse or anti-rabbit horseradish peroxidase conjugated antibodies (Bio-Rad, Hercules, CA, USA) were also used. Protein–antibody complexes were revealed with a chemiluminescence detection kit (Super-Signal West Pico, Pierce, Waltham, MA, USA).

### 4.5. RNA Extraction and Northern Blotting Analyses

Total RNA was prepared and analysed by Northern blotting, exactly as described in [78]. Immunoprecipitation experiments were performed exactly as described in [78] using anti-HA agarose beads (EZview^TM^ Red Anti-HA Affinity Gel, Sigma-Aldrich, Burlington, MA, USA). The oligo probes used for Northern blotting were the following: 20S.3 (5′ TTAAGCGCAGGCCCGGCTGG 3′), 23S.1 (5′ GATTGCTCGAATGCCCAAAG 3′), and rRNA2.1 (5′ GGCCAGCAATTTCAAGTTA 3’).

### 4.6. Microscopy

To study the nuclear localisation of Dbp7, the strain YKL500 [62], which expresses the red fluorescent protein yEmCherry fused to the nucleolar protein Nop58, was transformed with YCplac22-derived plasmids (see Appendix A) containing different constructs fused to a 3xyEGFP tag. Cells were visualised using an Olympus BX61 fluorescence microscope equipped with a digital camera. Images were analysed using the Cell Sens software (Olympus, Tokio, Japan) and processed with Adobe Photoshop CC 2017 (Adobe Systems Inc., San José, CA, USA).

### 4.7. Sucrose Gradient Centrifugation

Cell extracts for polysome and r-subunit analyses were prepared as previously described [79], using an ISCO UA-6 system equipped to continuously monitor A_254_. When required, fractions of 0.5 mL were collected from the gradients, and proteins were extracted from each fraction [80] and analysed by Western blotting.

### 4.8. Reproducibility

Unless otherwise specified, experiments were conducted at least three times (independent biological replicates) with at least two (normally three) technical replicates to give a sum of at least six to nine recordings, respectively. In figures, only a representative result is shown; the rest of the independent experiments showed the same results.

## Figures and Tables

**Figure 1 ijms-24-03460-f001:**
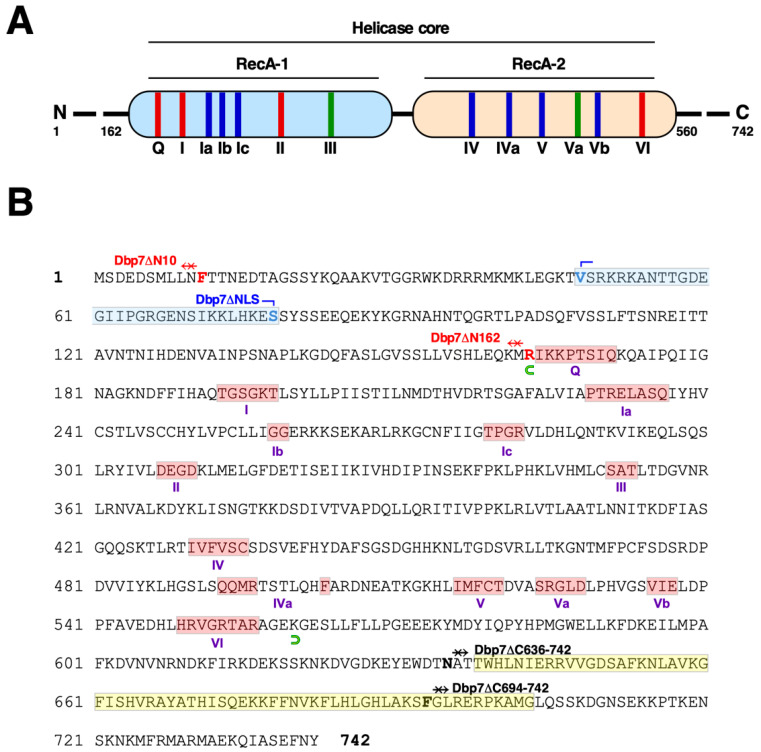
Dbp7 is a DExD-box protein. (**A**) Schematic representation (not to scale) of the primary amino acid sequence of Dbp7. The figure shows the helicase core composed of two RecA-like domains and the flanking N-terminal and C-terminal extensions. The length of each extension is depicted by numbers. The different conserved sequence motifs found in RNA helicases are also shown. They are coloured according to their predominant biochemical function: red, ATP binding and hydrolysis; blue, nucleic acid binding; green, coupling between nucleic acid and ATP binding sites. (**B**) Amino acid sequence of Dbp7. Motifs are highlighted in a pale red colour; a putative nuclear localization signal is highlighted in pale blue; and the C-terminal DUF4217 domain of unknown function in pale yellow. The conserved motifs are defined according to [9]. The helicase core region between the end of the N-terminal extension (M162) and the beginning of the C-terminal extension (K560) is delimited by green half ellipse symbols. The different mutations described in this work are also shown.

**Figure 2 ijms-24-03460-f002:**
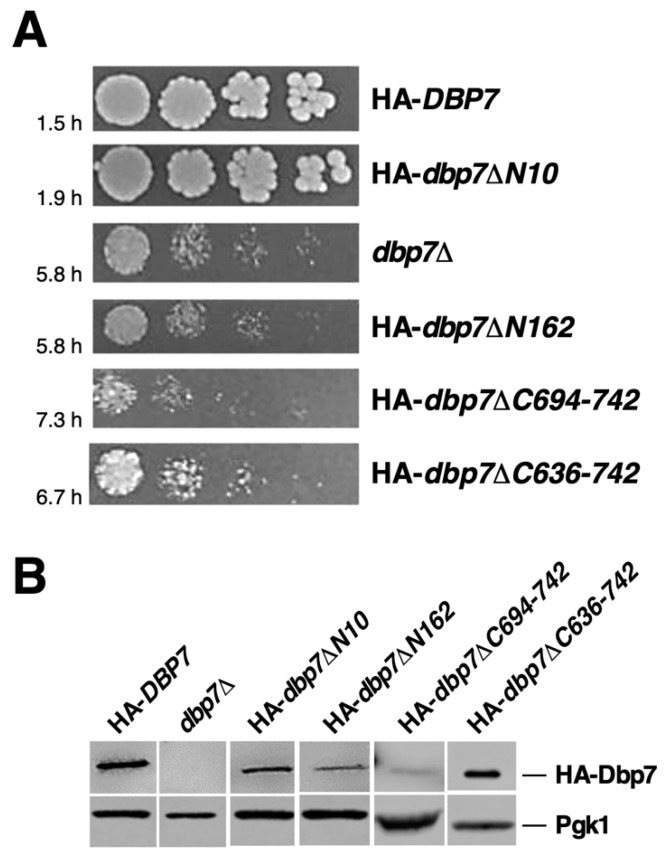
Growth phenotype of the truncated *dbp7* mutants at the N- and C-terminal extensions. (**A**) Growth analysis of the indicated strains compared to their isogenic wild-type control. Strain JuCY1 was transformed with different plasmid-borne *DBP7* alleles: HA-*DBP7* (wild-type control), HA-*dbp7∆N10*, HA-*dbp7∆N162*, HA-*dbp7∆C694-742*, and HA-*dbp7∆C636-742*. A control transformed with an empty plasmid (*dbp7∆*) was also used. Strains were 5-fold serially diluted and spotted on SD-Trp plates, which were incubated at 30 °C for 3 days. Doubling times of the different strains grown in liquid SD-Trp medium at 30 °C are indicated. (**B**) The indicated strains were grown in liquid SD-Trp medium and harvested at an OD_600_ of 0.8; whole cell extracts were prepared and equivalent amounts of the different cell extracts were subjected to Western blotting analyses with antibodies against the HA epitope and Pgk1.

**Figure 3 ijms-24-03460-f003:**
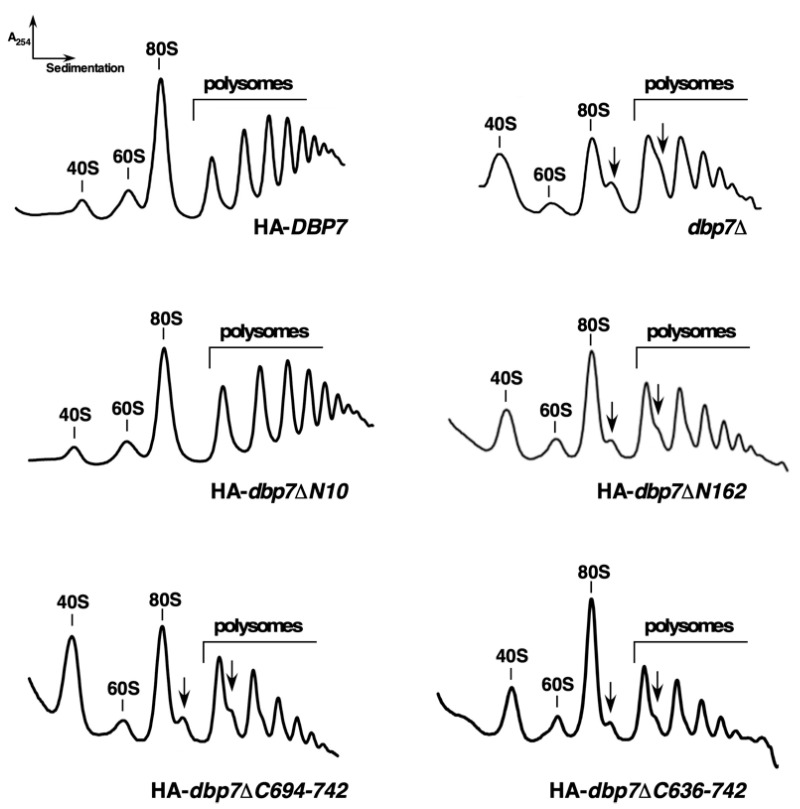
The N- and C-terminal truncations of Dbp7 induce a deficit in 60S r-subunits. Strains described in the legend of Figure 2 were grown to mid-log phase in liquid SD-Trp medium at 30 °C. Cell extracts were prepared, and 10 A_260_ units were resolved on 7–50% sucrose gradients. The A_254_ was continuously measured during gradient fractionation. Sedimentation is from left to right. The peaks of free 40S and 60S r-subunits, 80S free couples/monosomes, and polysomes are indicated. Half-mers are indicated by arrows.

**Figure 4 ijms-24-03460-f004:**
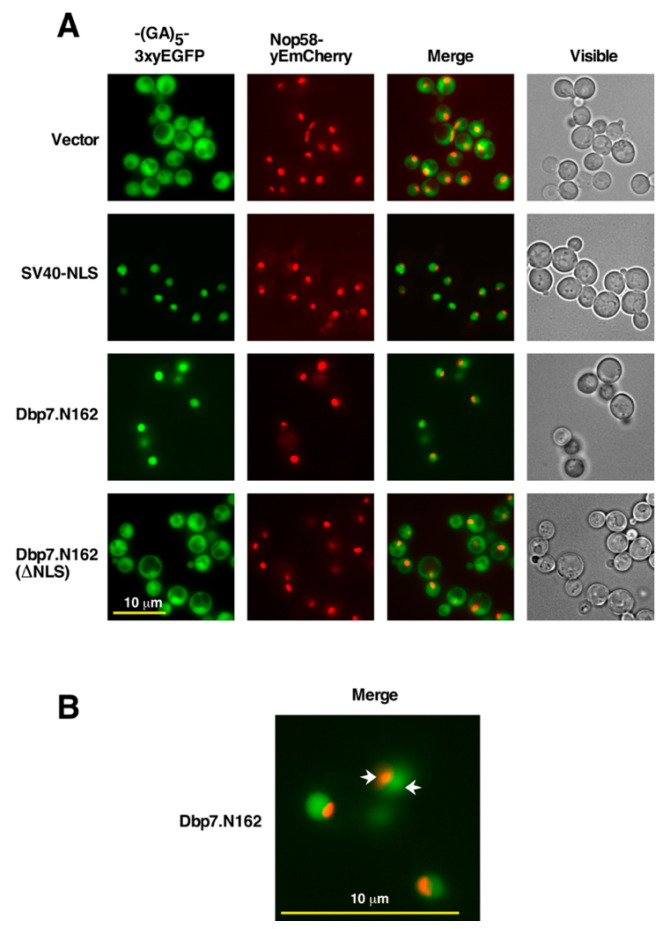
The N-terminal extension of Dbp7 contains a functional NLS sequence. (**A**) Strain YKL500, which expresses a yEmCherry-tagged Nop58, was transformed with plasmids expressing from the *ADH1* promoter (i) a (GA)_5_-3xyEGFP reporter alone (Vector). (ii) The NLS sequence of the SV40 large T-antigen fused to the (GA)_5_-3xyEGFP reporter (SV40-NLS). (iii) The first 162 amino acids of Dbp7 fused in frame to the (GA)_5_-3xyEGFP reporter (Dbp7.N162). (iv) The previous fragment, but lacking the segment from V48 to S78 (Dbp7.N162(∆NLS)). After transformation, cells were grown in liquid SD-Leu medium at 30 °C and the localisation of the GFP fusions was determined by fluorescence microscopy. Nucleoli were revealed by the nucleolar marker protein Nop58-yEmCherry. Cells were identified under bright field illumination (Visible). Approximately 200 cells were examined for each reporter, and practically all cells gave the same results as the selected cells shown in the pictures. (**B**) Magnified picture of selected cells expressing the Dbp7.N162 construct fused in frame to the (GA)_5_-3xyEGFP reporter. Note that the signal does not colocalise with the nucleolus, as was revealed by the Nop58-yEmCherry. The arrows pinpoint the nucleolus (red) and the nucleoplasm (green).

**Figure 5 ijms-24-03460-f005:**
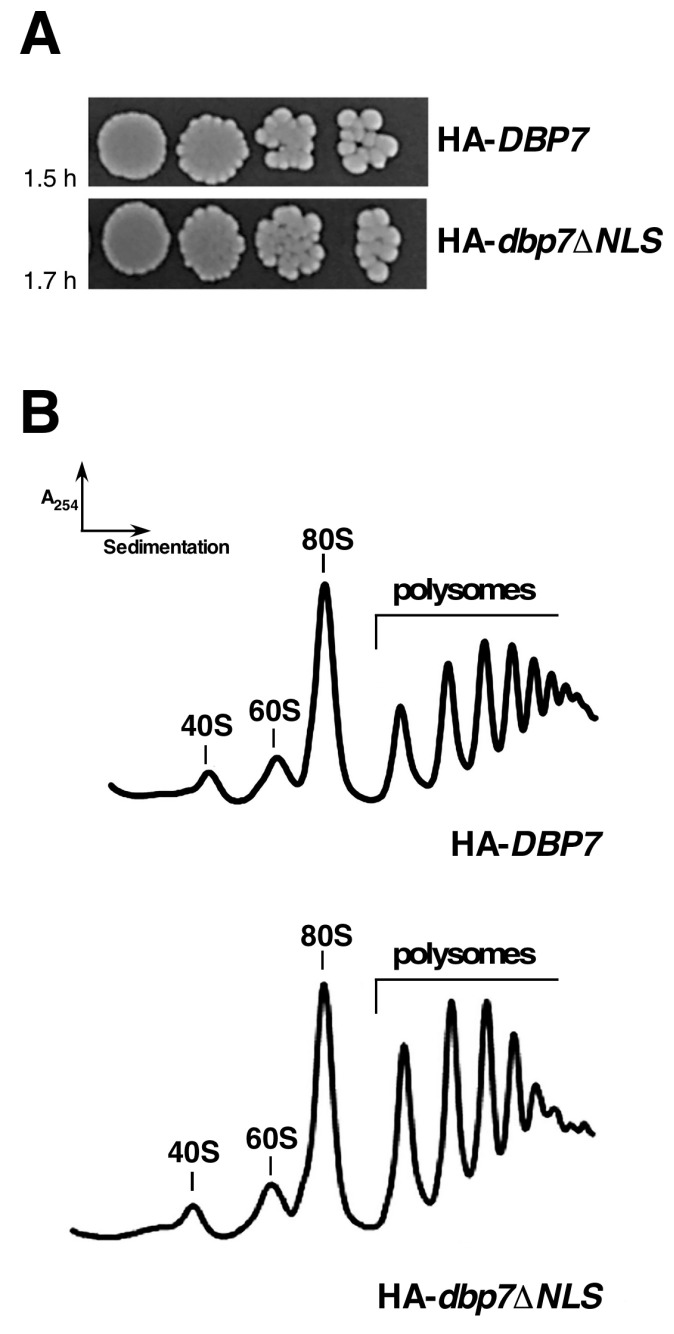
The HA-*dbp7∆NLS* mutation slightly reduces growth but does not affect polysome profiles. (**A**) Growth of the mutant harbouring the HA-*dbp7∆NLS* allele as the sole source of Dbp7. Strain JuCY1 was transformed with a plasmid harbouring either the HA-*DBP7* or the HA-*dbp7∆NLS* allele. Cells were grown in a liquid SD-Trp medium, and serial 5-fold dilutions were spotted onto SD-Trp plates, which were incubated at 30 °C for 3 days. The doubling times of the strains grown in liquid SD-Trp medium at 30 °C are indicated. (**B**) The above strains were grown in liquid SD-Trp to an OD_600_ of 0.8. Cell extracts were prepared and 10 A_260_ units of each extract were resolved on 7–50% sucrose gradients. The A_254_ was continuously measured and recorded during gradient fractionation. Sedimentation is from left to right. The peaks of free 40S and 60S r-subunits, 80S free couples/monosomes, and polysomes are indicated.

**Figure 6 ijms-24-03460-f006:**
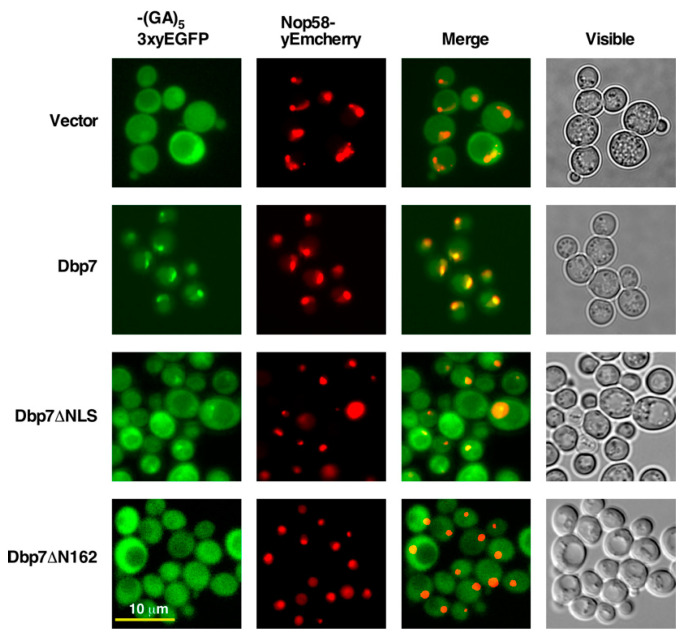
Dbp7 lacking the N-terminal extension fails to accumulate in the nucleus. Strain YKL500, which expressed a yEmCherry-tagged Nop58, was transformed with different plasmids harbouring the (GA)_5_-3xyEGFP reporter: (i) the reporter alone (Vector), (ii) a full-length Dbp7 fused to the reporter at its C-terminal end (Dbp7), (iii) a Dbp7 variant protein lacking the 31 amino acids (V48 to S78) corresponding to the NLS (Dbp7∆NLS) fused to the reporter at its C-terminal end, and (iv) a Dbp7 variant protein lacking the first 162 amino acids from its N-terminal extension fused to the reporter at its C-terminal end (Dbp7∆N162). Transformants were grown in SD-Leu medium at 30 °C, and the localisation of the GFP-fused proteins was studied by fluorescence microscopy. The nucleolus was revealed by the nucleolar marker protein Nop58-yEmCherry. Cells were identified under bright field illumination (Visible). Approximately 200 cells were examined for each reporter, and practically all cells gave the same results as those selected and shown in the pictures.

**Figure 7 ijms-24-03460-f007:**
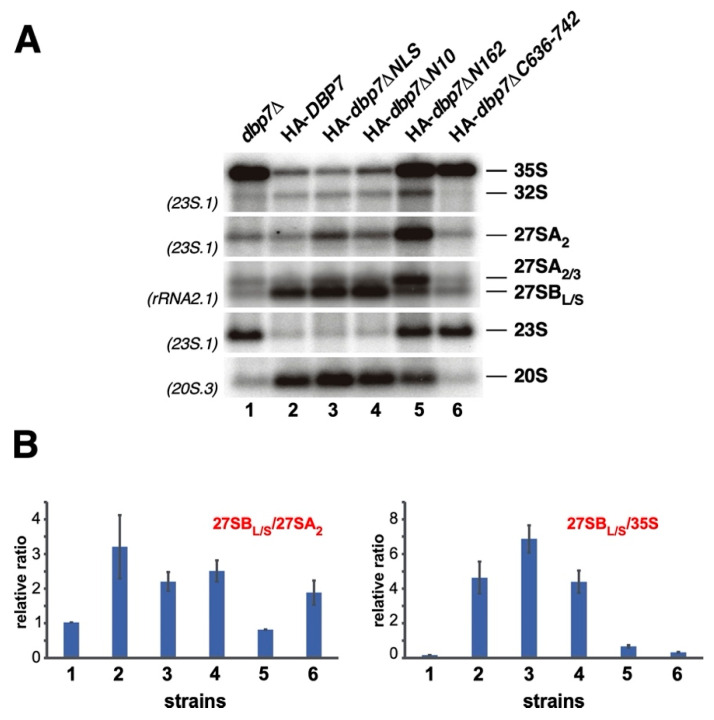
The N- and C- terminal truncations of Dbp7, as well as the lack of its NLS, impair pre-rRNA processing. (**A**) Strain JuCY1 transformed with an empty vector (*dbp7∆*) or different plasmid-borne *DBP7* alleles: HA-*DBP7*, HA-*dbp7∆NLS*, HA-*dbp7∆N10*, HA-*dbp7∆N162,* and HA-*dbp7∆C636-742* were grown in liquid SD-Trp medium at 30 °C and harvested at an OD_600_ of 0.8; total RNA was extracted and ca. 4 μg were used for Northern blotting. The membrane was hybridized consecutively with probes 23S.1, rRNA2.1, and 20S.3, allowing the detection of the indicated pre-rRNA species. (**B**) Quantification of the 27SB_L/S_/27SA_2_ and 27SB_L/S_/35S precursor ratios from two independent phosphorImager data using the MultiGauge software.

**Figure 8 ijms-24-03460-f008:**
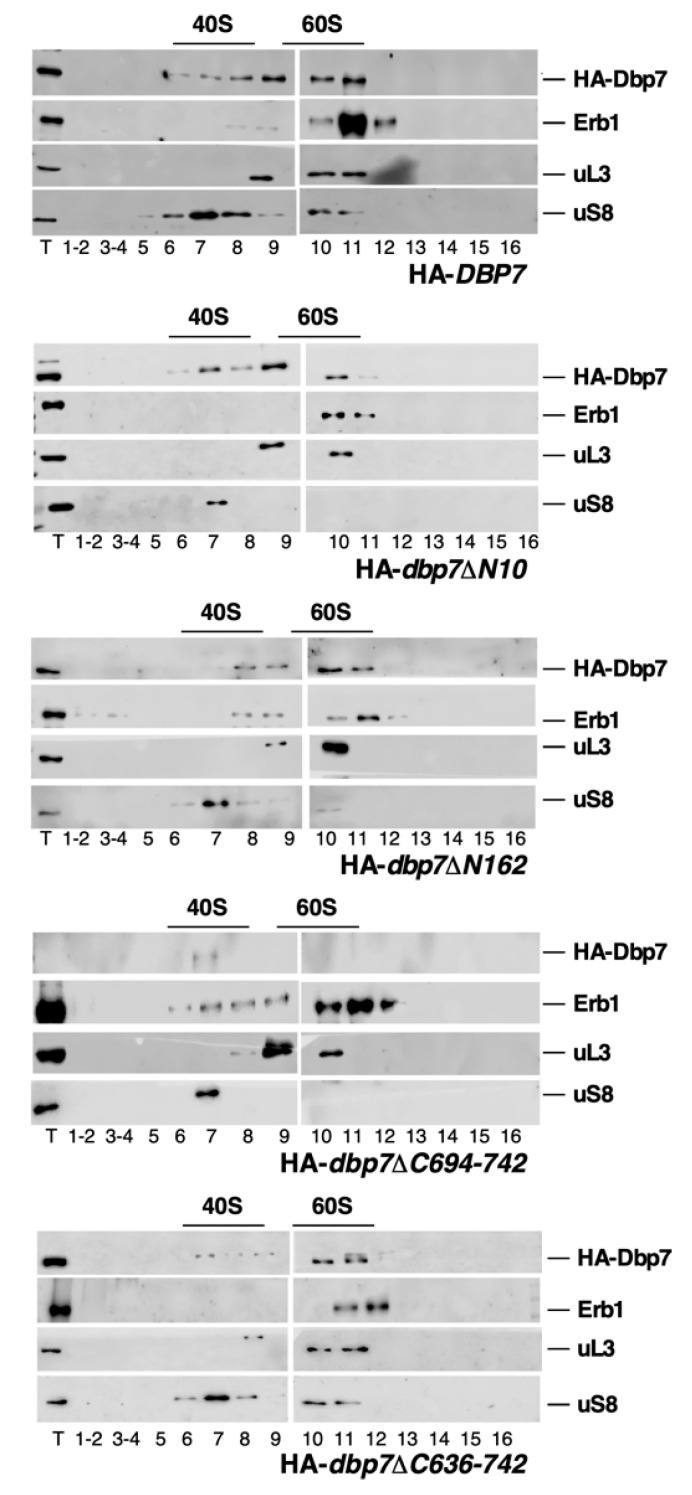
Truncation of either the N- or C-terminal extension of Dbp7 does not impair its co-sedimentation with pre-60S r-particles on gradients. Total cell extracts were prepared from the indicated strains, which express as sole source of Dbp7: wild-type full-length HA-Dbp7, HA-Dbp7∆N10, HA-Dbp7∆N162, HA-Dbp7∆C694-742, and HA-Dbp7∆C636-742. All strains were grown to the exponential phase in liquid SD-Trp medium at 30 °C. Then, 10 A_260_ units of each extract were resolved on 7–50% sucrose gradients at a low concentration of Mg^2+^ to dissociate ribosomes into r-subunits. Sedimentation is from left to right. The sedimentation positions of 40S and 60S r-subunits are indicated. Fractions (numbered 1–16 below each panel) were collected from the gradients; proteins were extracted from the same volume of each fraction and subjected to Western blot analysis, with specific antibodies detecting the HA epitope of tagged Dbp7 and the proteins Erb1, uL3, and uS8.

**Figure 9 ijms-24-03460-f009:**
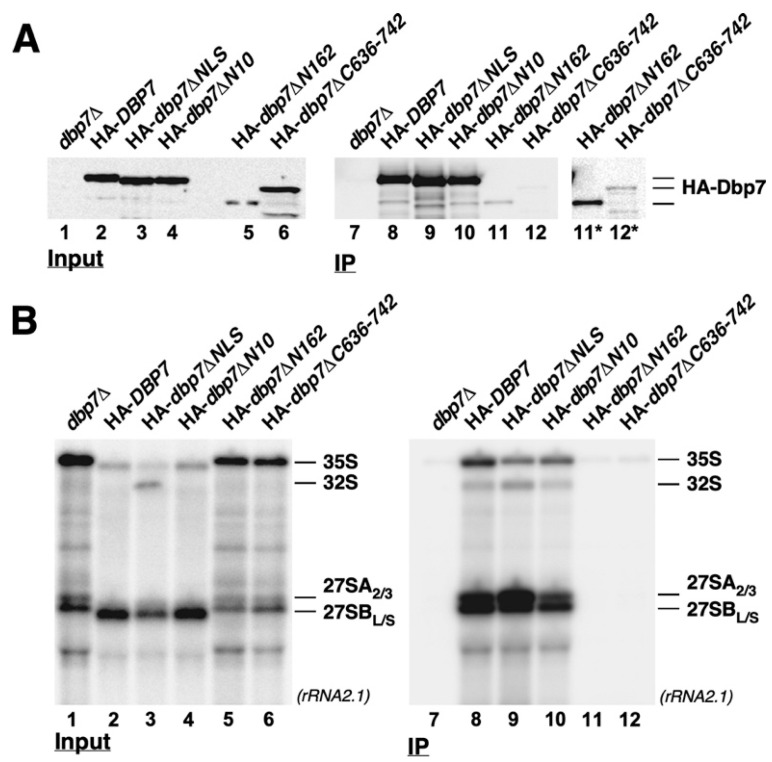
Truncation of either the N- or C-terminal extension of Dbp7 impairs the co-immunoprecipitation of pre-ribosomal particles. (**A**) Strain JuCY1 transformed with an empty vector (*dbp7∆*) or different plasmid-borne *DBP7* alleles: HA-*DBP7*, HA-*dbp7∆NLS*, HA-*dbp7∆N10*, HA-*dbp7∆N162,* and HA-*dbp7∆C636-742* were grown in liquid SD-Trp medium at 30 °C and harvested at an OD_600_ of 0.8. Total cell extracts were prepared for each strain, and the complexes associated with the different versions of the HA-Dbp7 protein were affinity purified using anti-HA agarose beads. (**A**) Total protein was extracted from aliquots of input extracts (Input) or immunoprecipitated samples (IP) and analysed by Western blotting. The HA-tagged Dbp7 proteins were detected using anti-HA antibodies. Note that in input lanes 5 and 6, the amount of protein loaded was increased 6-fold relative to input lanes 1–4. Lanes 11* and 12* correspond to an increased image acquisition time of lanes 11 and 12. (**B**) Total RNA was extracted from the same samples and subjected to Northern blotting. Pre-rRNAs were detected using the rRNA2.1 probe.

## Data Availability

Not applicable.

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
