# Peer review of "The Terminal Extensions of Dbp7 Influence Growth and 60S Ribosomal Subunit Biogenesis in Saccharomyces cerevisiae"

_ijms, 2023, doi:10.3390/ijms24043460_

Round 1
Reviewer 1 Report (Previous Reviewer 1)
The manuscript is now acceptable for publication. The authors have nicely addressed all of my concerns.
Author Response
Thanks
Reviewer 2 Report (New Reviewer)
The article titled “The terminal extensions of Dbp7 influence growth and 60S ri- bosomal subunit biogenesis in Saccharomyces cerevisiae” by Contreras et al., build on their earlier work on Dbp7. Here the authors try to show that the N and C-terminal extensions of the Dbp7 Helicase are essential for optimal growth and essential for the biogenesis of 60S subunit of ribosome. Overall, it’s a good study. However, the authors need to address the following concerns before accepting this piece of work for publication in IJMS.
Concerns:
1) Here is the Alphafold prediction which shows the predicted structure of Dbp7: Regions colored in Blue is the N-terminal (1-162), Green is the core Helicase domain (162-635) and Orange is the C-terminal domain (636-742). The Extensions look like they stabilize the Core domain especially the C-terminal domain which is folding back to the Helicase domain. The authors need address this as deletion of the C-terminal may destabilize the Helicase activity and also the protein as evident from low expression levels. Maybe the authors can consider showing the invitro activity of helicase with these truncations.
2) Provide quantification for figure 3 and also inHA-dbp7dC694-742, the accumulation of 40S is here when compared to any other strain. Please explain.
3) Figure4: Can you quantify the localization using cellular fractionation? Or quantify the fluorescence and plot cytosolic to nucleus localization.
4) Figure 7: Is the experiment performed in triplicates? `please provide the statistical significance for the bar graph along with data points.

Author Response
Please see the attachment corresponding to the cover letter to the editor
Thanks for your work

Round 2
Reviewer 2 Report (New Reviewer)
It would be nice if the authors can show the in vitro helicase assay as it is very established in the lab.
Author Response
Please read the cover letter

Round 3
Reviewer 2 Report (New Reviewer)
Thank you for the response.
This manuscript is a resubmission of an earlier submission. The following is a list of the peer review reports and author responses from that submission.
Round 1
Reviewer 1 Report
This manuscript from Jesus de la Cruz, Yves Henry and Anthony Henras addresses an important question in RNA biology: what is the role of the N- and C-terminal extensions of DEAD box RNA helicases? The conserved recA domains of these proteins are not thought to recognize specific RNA sequences. Thus, these enzymes are likely directed to their specific RNA or RNP substrates, and perhaps even regulated, via interactions with their unique N- and/or C-terminal sequences. The precise function of DEAD box helicase extensions has only been explored in a very few cases.
Here, the authors investigate effects of truncating, mutating or completely deleting each extension of Dbp7 involved in large ribosomal subunit assembly in yeast. The bottom line is that both extensions are important for the function of Dbp7. However, surprisingly, neither extension is necessary for efficient association of Dbp7 with pre-ribosomes, as assayed by co-sedimentation on sucrose gradients. In particular, the N-terminal extension contains an NLS, but experiments suggest the presence of a second means for some fraction of Dbp7 to enter nuclei.
To improve this manuscript, I suggest that the authors consider the following issues:
(1) Figure 1: Please enlarge this figure; it is difficult to see all of the necessary detail.
Why did the authors designate the end of the N terminal extension and the beginning of the C terminal extension as shown? What structures are predicted by alpha-fold, for example, beyond these endpoints? Might such structures be conserved among DEAD box proteins and be part of the recA domains?...or not? It appears that the authors consider the extensions to begin or end immediately adjacent to previously described conserved sequences involved in binding to ATP or RNA.
How conserved are the extensions among all Dbp7 proteins?
What might be the function of this DUF domain? Is it, or something resembling it, present in other RNA helicases beyond those listed in the manuscript?
(2) The N-terminal extension clearly is important for Dbp7 function. Its deletion significantly decreases the stability of Dbp7. Consistent with this result, deletion of the N terminal extension creates phenotypes like that of a full deletion of the gene. This extension harbors a bona fide NLS sufficient for nuclear localization (Figure 4). However, deletion of only this NLS does not compromise growth, stability of the protein, or amounts of 60S subunits (Figure 5), but does result in partial nuclear localization of Dbp7 (Figure 6). This leads the authors to reasonably suggest that there is a second, weak NLS elsewhere in Dbp7, or else that Dbp7 piggybacks into the nucleus. Did the authors’ computational screen for NLS’s reveal a weak fit elsewhere? Interestingly, all of the careful analysis of the N-terminal extension suggests that it might have a function other than harboring the one identified NLS and other than directing Dbp7 to pre-ribosomes (but see below).
(3) Deletions of the C-terminal extension yielded more curious results: deleting the last 49 amino acids had a stronger effect on growth and Dbp7 stability (but a similar effect on 60S subunits ) cf. deleting the last 107 amino acids. Can the authors speculate why this is the case? It seems most likely that this reflects ability of the lesser truncated protein to assemble into pre-ribosomes and perturb their assembly. The authors tested this idea by testing whether these deletions exhibit dominant negative phenotypes, and they did not. Perhaps the Dbp7 N693 deletion can assemble into pre-ribosomes and perturb assembly more than the N635 deletion, in the absence of competing wildtype protein, but neither can compete with wildtype protein to assemble in the authors’ dominant negative experiment. One scenario, in both cases, neither mutant protein that does not assemble might be turned over. However, in addition, the lesser stability of the 693 deletion Dbp7 cf. the 635 deletion Dbp7 might reflect turnover of aberrantly assembled pre-ribosomes containing that 693 deletion Dbp7, and thus additional turnover of that protein.
(4) By the way, why are the C-terminal deletion mutants labeled N635 and N693? Perhaps more clear: dbp7deltaC 635-742 and dbp7deltaC 693-742?
(5) Figure 7: Wild-type Dbp7 co-sediments with pre-60S ribosomes, although some is detectable in lighter fractions, but none in fractions where monomeric Dbp7 should sediment. Please speculate what these lighter fractions might mean…a subcomplex perhaps?
Interestingly and somewhat surprisingly, deleting neither extension affected co-sedimentation of Dbp7 with pre-ribosomes. However, from these experiments it is impossible to determine whether some fraction of these mutant Dbp9 proteins fails to assemble and is degraded. The nature of this beast: without an internal control, namely some unassembled Dbp7 sedimenting at the top of the gradient in wild-type cells, which is not observed, western blots of gradient fractions from wild-type vs mutant strains cannot quantify the relative efficiency of assembly of wildtype Dbp7 vs mutant Dbp7.
Furthermore, co-sedimentation does not prove that Dbp7 is present in pre-60S particles. This should be evaluated by affinity purifying pre-60S ribosomes.
Sincerely,
John Woolford
Reviewer 2 Report
Manuscript ID: ijms-1821951
Title: The terminal extensions of Dbp7 influence growth and 60S ribosomal subunit biogenesis in Saccharomyces cerevisiae
Contreras et al. analyze the functional significance of the N-terminal and C-terminal extensions of the RNA helicase Dbp7 in the model organism Saccharomyces cerevisiae. The authors subjected strains with Dbp7 knock out and N-terminal and C-terminal truncations to growth-, biochemical- and fluorescence microscopic analysis. Truncations in N- and C-terminus result in slow cell growth, and so-called half-mer formation, and a lack of nuclear localization.
Major points:
1) Altogether, the results are convincing and attractively presented. However, the quality of the western blots in Fig. 7 and S4 leaves much to be desired, which makes their interpretation difficult. It is possible that the problem occurred at the level of protein transfer, membrane blocking, primary or secondary stain, or a combination of all these. In addition, the parts of the blots in Fig. 7 are poorly aligned, both vertically and horizontally. Figure S 4 is really unacceptable. The amount of the loading control varies a lot and the corresponding GFP stain is not discernible. Western blots in Figs. 7 and S4 definitely should be repeated and if possible the individual samples of Fig. 7 should be loaded on one gel (instead of two), resulting in one blot membrane per mutant per stain (Ha, Erb1, uL3, uS8). This avoids problems with the vertical alignment and facilitates a homogeneous stain.
2) Figure 4A and B: Dbp7.N162 attains nuclear localization, but –according to the authors- does not interact with pre-60S. The fluorescence microscopy analysis is not entirely convincing. Analysis of this construct in a Western Blot (like Fig. 7) will unambiguously demonstrate, whether it has the ability to interact with 60S or pre-60S subunits.
Minor points:
1) A list of cartoons with all the generated Dbp7 constructs would be extremely helpful and could be implemented in Fig. 1A or 1B.
2) Line 152-153: The nomenclature is irritating. The C-terminal truncations should rather be named HA-dbp7DC693 and HA-dbp7DC635, or similar.
3) Line 188: The authors should explain what half-mers are and add a literature reference.
4) Line 253: Note that the signal does not colocalised with the nucleolus… . The sentence should read: … signal does/ did not colocalise…, …. signal is not colocalising…, or similar.
5) Line 264 ff.: What does this mean? The predicted NLS is no NLS. How reliable are these predictions? Are there other examples from literature?
6) Line 281: Why is a triple yEGFP fusion used? Wouldn´t a single yEGFP fusion be sufficient?
7) Line 365-367: This conclusion cannot be drawn. Once membranes of the cell and its compartments are disrupted, all the molecules that used to be separated have the chance to interact with each other and form complexes stable enough to survive sucrose gradient ultra-centrifugation. Rather a conclusion could be that Dbp7 constructs, even though lacking N- and C-terminal extensions with putative sequences permitting nuclear localization, have the capability to interact with 60S- or pre-60S subunits.
8) The authors should expand their discussion and might find some of the following thoughts helpful.
- The N-terminus of Dbp7 is predicted to be unstructured and doesn´t seem to interact with the compactly folded body of the protein. From a structural standpoint, a deletion of the first 150 amino acids should not interfere with the overall folding of the protein. However, according to the AlphaFold prediction the C-terminus is mostly alpha-helical and contributes to the architecture of the RecA-2 domain. Hence, C-terminal deletions could interfere with protein folding, stability and therefore abundance.
- The predicted or known nucleic acid binding sites are located in the RecA-like domains.
- N- and C-terminal truncations still can interact with 60S or pre-60S subunits, but either can´t enter the nucleus, or are lower concentrated. Both scenarios could result in suboptimal 60S biogenesis and explain both the occurrence of half-mers and the slow growth phenotypes.